# Comparison of the Prevalence of Eating Disorders among Dietetics Students and Students of Other Fields of Study at Selected Universities (Silesia, Poland)

**DOI:** 10.3390/nu14153210

**Published:** 2022-08-05

**Authors:** Aneta Matusik, Mateusz Grajek, Patryk Szlacheta, Ilona Korzonek-Szlacheta

**Affiliations:** 1Department of Prevention of Metabolic Diseases, Faculty of Health Sciences in Bytom, Medical University of Silesia in Katowice, 41902 Bytom, Poland; 2Department of Public Health, Faculty of Health Sciences in Bytom, Medical University of Silesia in Katowice, 41902 Bytom, Poland; 3Department of Toxicology and Health Protection, Faculty of Health Sciences in Bytom, Medical University of Silesia in Katowice, 41902 Bytom, Poland

**Keywords:** eating disorders, EAT-26, students, dietetics

## Abstract

Background: Over the past few years, an increase in the incidence of eating disorders has been noted. An increase in the pace of life, an increase in the availability of a wide variety of food products, and, to a large extent, the involvement of mass media are cited as reasons for this phenomenon. The promotion of a slim figure by the mass media is equated with achieving success in life, but also the advertising of a wide selection of food products (often highly processed) can have a serious impact on the development of eating disorders. This phenomenon is particularly observed in industrialized Western countries. Objective: Therefore, it was decided to test and compare whether dietetics students are indeed more predisposed to developing eating disorders than students not in the nutrition field. Material and methods: the study included 310 individuals representing two equal groups of fields of study—dietetics and other students. The study used standardized questionnaire—EAT-26. Results: It was found that almost half (46%) of the respondents (both dietetics students and students of other majors) met at least one criterion out of three that could indicate the probable existence or susceptibility to an eating disorder. These individuals should see a specialist for further diagnosis. There was no significant effect of the field of study on the overall EAT-26 test score (*p* > 0.05). When this result was corrected for BMI values for those with the lowest scores on this indicator, the risk of eating disorders was found to be higher among students of majors other than dietetics (X^2^ = 13.572; V = 0.831 *p* = 0.001). Conclusions: Almost half of the respondents in both study groups showed a predisposition to eating disorders based on the EAT-26 test. Despite the presence of a correlation in individual responses that dietetics students are more predisposed to eating disorders, no such relationship was found according to the final EAT-26 test scores. However, it was observed that non-dietetics students who had low BMI values showed higher tendencies toward behaviors indicative of eating disorders.

## 1. Introduction

Eating disorders are disease entities with underlying psychological factors. They are defined as persistent behaviors associated with food intake leading to changes in consumption, contributing to psychosocial impairment and mental disorders [1,2]. People who struggle with an eating disorder often experience depression. They often become addicted to alcohol, drugs, or sexual activities, for example, and self-harm. The symptoms of eating disorders are closely related to the internal feelings of the patient (e.g., pain, stress, fear, loneliness, low self-esteem). The literature distinguishes several factors that can affect the formation of an abnormal relationship with food. The most commonly cited are psychological, biological, social, behavioral, and cultural factors [3]. In particular, the age of the sufferer may be linked to factors that influenced the development of the disorder. Among the young population, family and environmental factors come first. Among the causes of an appetite disorder in children and adolescents are body fat content in girls, hormonal changes, and the influence of peer groups. In adults, socio-cultural factors can influence the disorder [4]. Appetite disorders have been differentiated according to the criteria outlined in the major mental illness classification systems: International Statistical Classification of Diseases and Related Health Problems ICD-11 in European countries and the American Psychiatric Association’s Diagnostic and Statistical Manual of Mental Disorders DSM-5 in the United States [5].

Over the past few years, an increase in the incidence of eating disorders has been noted, yet Poland still has very few epidemiological studies on eating disorders. The most reliable data from 2011 indicate that anorexia nervosa occurs in about 2% of people before the age of 18, and these are 10 times more likely to be female. There are no studies in Poland estimating these values in terms of the elderly population and people belonging to specific groups (physically active people, associated with healthy eating and attention to physical appearance) [6]. However, an increase in the pace of life, an increase in the availability of a wide variety of food products, and, to a large extent, the involvement of mass media are cited as reasons for this phenomenon. The promotion of a slim figure by the mass media is equated with achieving success in life, but also the advertising of a wide selection of food products (often highly processed) can have a serious impact on the development of eating disorders. This phenomenon is particularly observed in industrialized Western countries. Eating disorders mainly (but not exclusively) affect adolescents and young adults. An increase in the incidence of an abnormal relationship with food, especially among people who practice endurance and aesthetic sports, such as running, ballet, gymnastics, or figure skating, has also been observed. What is more, an increase in the prevalence of eating disorders has been noted among those studying courses related to proper nutrition [7,8]. Dietetics students, by their curriculum, are subjected to intensive education on proper nutrition. In addition to the knowledge they acquire in classes, they increase their knowledge of contemporary fashionable diets and nutritional trends. In this way, they want to meet the expectations set by their future patients. They begin to program their lives to look and behave healthily throughout lives, which can in turn trigger a nascent obsession with nutrition and their figure. Society’s pressure for them to be impeccable role models in eating and appearance may be the cause of their developing an abnormal relationship with food [9].

Since there is a lack of research on this issue in recent years, especially in Poland, it was decided to check in an in-house study whether dietetics students are indeed more likely to suffer from eating disorders than students of other majors. The group studied in the in-house paper consisted of 310 students: 155 students in dietetics and 155 students in other majors unrelated to nutrition.

Therefore, it was decided to test and compare whether dietetics students are indeed more predisposed to developing eating disorders than students not in the nutrition field. The basis of the study was to address the hypothesis that the prevalence of eating disorders among dietetics students is higher than among students in other fields of study.

## 2. Materials and Methods

### 2.1. Study Area and Sample

The survey was conducted from March to April 2022 among dietetics students. The survey was conducted using a mixed survey method, and a questionnaire technique. In this study group, 155 dietetics students participated in the direct survey, including 96% women and 4% men. The survey was also conducted among students of other majors of the selected universities by electronic means, as an indirect survey (CAWI). This group also accounted for 155 students, including 77% women and 23% men. Students of an economic university comprised 51 respondents, students of a general university comprised 52 respondents, and students of a music university comprised 52 respondents. A total of 310 students between the ages of 18 and 25 participated in the research work. They were students at different stages of their studies.

### 2.2. Research Tool

The prevalence of eating disorders was analyzed using the EAT-26 questionnaire and a metric. The metric included gender, age, university name, and degree. In addition, students were asked about their current body weight and height. Based on this, from the formula: body weight (kg)/height (m^2^), BMI (kg/m^2^) was calculated, which was interpreted according to the WHO-approved BMI classification for adults [10] (Table 1).

The research study used an eating disorder screening tool, the American Eating Attitudes Test (EAT-26) questionnaire by Garner et al. [11]. The EAT-26 is a standardized questionnaire for detecting risk symptoms of eating disorders. It is designed both for screening individuals with a clinical diagnosis and for screening among those at risk for anorexia, bulimia, or obesity. The EAT-26 is one of the most widely used screening tools in eating disorder prevalence studies worldwide. The test is an abbreviated version of the EAT-40 created by Garner et al. [11] The interpretation of the EAT -26 questionnaire consists of three “referral criteria” that determine whether the respondent should come in for a further assessment of an eating disorder risk. They include:(1)EAT total actual score consists of 26 questions or statements on attitudes toward nutrition. Items 1–25 are scored as follows: Always = 3; Usually = 2; Often = 1; Other answers = 0. Item 26 is scored in reverse (Never = 3, etc.) The screening test can be scored from 0 to 78. A respondent with a score of 20 or more is at risk of developing an eating disorder and should see a specialist for further diagnosis.(2)Behavioral questions indicate possible symptoms of an appetite disorder or recent significant weight loss. They concern compensatory behaviors (use of laxatives, weight loss, provoking vomiting, overeating, engaging in excessive physical activity, and significant weight loss in a short period). If the respondent answered affirmatively to any of the behavioral questions, as shown in the table, this may indicate the existence of abnormalities and the need for further diagnosis of eating disorders (Table 2).

(3)Low body weight compared to age norms: the questionnaire includes detailed questions about height, weight, and gender. This information was used to calculate the body mass index (BMI) to determine the possible risk of an eating disorder. Below is a table that indicates whether the subject is underweight by age and gender (Table 3).

### 2.3. Eligibility Criteria and Ethical Consent

The criteria for inclusion in the study group were the following two conditions: (1) voluntary participation in the study and complete completion of the questionnaire and (2) student status at the time of the study. In addition, during the initial interview with the respondents, they were asked about previous psychiatric episodes, thus checking their history of using a psychologist or psychiatrist. The fact, of their use was noted as a criterion for exclusion from further study, as those with prior diagnosis and treatment of eating disorders, mood disorders, anxiety disorders, and others could significantly affect the final outcome of the study.

All study participants gave informed consent to participate in the study by completing a questionnaire. The study was approved by the Bioethics Committee of the Medical University of Silesia in Katowice (PCN/0022/KB/211/20) in light of the Act on Medical and Dental Professions (5 December 1996), which includes a definition of medical experimentation. The study participants consciously agreed to participate in the study.

### 2.4. Statistical Analysis

Statistical analysis was performed using Statistica 13.0. The analysis was performed via the Chi-square test. A *p* = 0.05 was taken as the level of statistical significance. The V-Cramer correlation coefficient was also used to test the strength of the relationship of statistical characteristics. In each case where the symbol NS (non-significance) was placed next to the *p*-value, it indicates a lack of statistical significance.

## 3. Results

### 3.1. Sample Characteristics

A total of 310 students participated in the conducted study. The first half of the group (*n* = 155) consisted of medical university dietetics students. The second half of the study group consisted of students from various majors of selected non-dietetics universities (*n* = 155). In total, 51 students participated from the economic university, 52 students participated from the general university, and 52 students participated from the music university. Among the dietetics students surveyed, women accounted for 96% (*n* = 149) and men for 4% (*n* = 6) of the respondents. However, in the second group of respondents, women accounted for 77% (*n* = 120) and men for 23% (*n* = 35) of the respondents. Among dietetics students, 68% of respondents declared that they were in their bachelor’s degree program, and 32% of respondents were in their second-degree program. In the second group, 62% of respondents were first-degree students, while 38% were second-degree students.

### 3.2. BMI of Participants

The calculated BMI of the dietetics students showed that 12% (*n* = 19) of the respondents were overweight, 2% (*n* = 3) were first-degree obese, 8% (*n* = 13) were underweight, and 2% (*n* = 3) were emaciated. Body weight in the normal range was 75% (*n* = 117) of dietetics students. In contrast, among students in other majors, 15% (*n* = 23) of respondents were overweight, 5% (*n* = 8) were first-degree obese, 8% (*n* = 12) were underweight, 4% (*n* = 6) were emaciated, and 1% (*n* = 1) were starved. Normal body weight was 68% (*n* = 105) of the students. Based on the calculated BMI of the respondents and subsequent comparison with age norms, it was found that 14% of dietetics students and 17% of students in other majors had too low body weight compared to age norms. There was no significant effect of the field of study on low body weight compared to age norms (*p* > 0.05-NS).

### 3.3. Risk of Eating Disorders

Based on the assigned score from the EAT-26, Part A questionnaire, it was estimated that 15% of both dietetics students and students from other majors are at risk for eating disorder-related diseases and should seek evaluation by a specialist for further diagnosis. There was no significant effect of the field of study on the total score of the actual items of the EAT-26, Part A test (≥20), which may indicate the risk of developing an eating disorder (*p* > 0.05-NS). According to the accepted scores on behavioral questions from the EAT-26 test, Part B, it was estimated that 33% of dietetics students and 28% of students in other majors met a criterion that could indicate a risk of developing an eating disorder. There was no significant effect of the field of study on the EAT-26 test score on behavioral questions (*p* > 0.05-NS). Based on the overall scores and interpretation of the EAT-26 questionnaire, it was found that almost half (46%) of the respondents (both dietetics students and students of other majors) met at least one criterion out of three that could indicate the probable existence or susceptibility to an eating disorder. These individuals should see a specialist for further diagnosis. There was no significant effect of the field of study on the overall EAT-26 test score (*p* > 0.05-NS). When this result was corrected for BMI values for those with the lowest scores on this indicator, the risk of eating disorders was found to be higher among students of majors other than dietetics (X^2^ = 13.572; V = 0.831 *p* = 0.001). All the results described are summarized in Table 4.

## 4. Discussion

Nowadays, eating disorders are a common phenomenon. Over the past 50 years, there has been an increase in the incidence of bulimia, anorexia, and compulsive eating syndrome in particular [12]. About 8–9% of the population has been found to struggle with gluttony or mental anorexia [13]. Craving disorders usually begin during adolescence and young adulthood. During this period, individuals are often prone to stressful events including taking exams, making decisions about the future, and moving to college. The result is an accompanying fear of adulthood. Studies have shown that the estimated prevalence of appetite disorders among college students ranges from 8 to 20% [14,15].

It is worrisome that a significant number of students who develop symptoms of eating disorders have not been diagnosed or sought treatment. Screening for this condition appears to be an important need to help sufferers in the early stages of the disease. Inappropriate dietary practices such as vomiting, fasting, restrictive diets, and laxative abuse, among others, can influence the development of disordered eating behavior. Nutritional counseling as part of a multidisciplinary approach plays a special role in the treatment of appetite disorders.

It may seem that dietetics students, thanks to specialized training in healthy eating habits, meal planning, or weight control, are less likely to have eating behavior disorders than those studying non-food-related majors. There is another belief that dietetics students view the start of their studies as a motivation to deal with their problems, which are inappropriate attitudes toward nutrition, as well as a desire to lower their body weight. These behaviors may exist before the start of the study but may also develop during education as a result of excessive preoccupation with healthy eating [14]. Our work confirms that 14% of nutrition students were overweight or obese. What is more, 14% of those surveyed also happened to lose 10 kg or more in the last 6 months. In contrast, among students of other majors, this percentage was only 5%; *p* < 0.05. In addition, 13% of dietetics students declared that they had been treated for eating disorders while, among students of other majors, this percentage was half as much at 6%; *p* < 0.05.

Another international study found that 77% of nutritionists from 14 countries believe that eating disorders are a problem for dietetics students [16]. Some studies suggest that the prevalence of eating disorders in nutrition students is higher than in students in other majors [17,18]. A study comparing eating behaviors between nutrition students and students in other majors in Portugal found that those in the first group showed greater dietary restrictions with subsequent bouts of overeating than students in other majors [17]. Another study conducted in South Africa also observed a higher risk of eating disorders among dietetics students 33.3% compared to other non-nutrition students 16.9%; *p* = 0.059 [19].

One important parameter in diagnosing bulimia or anorexia is an overestimation of body size. Sufferers perceive themselves as obese, despite being of the right weight or even underweight. Characteristically, there is a contradiction between BMI, actual images, and subjective assessment of the sufferer’s outward appearance [20]. Even though 85% of dieters were of normal weight or even underweight, as many as 32% of them “always”, “usually”, or “often” took steps to reduce their weight. In contrast, among students in other majors, 81% were normal weight, underweight, or skinny, and 32% of them chose to take action to lower their weight. The study by Buviora et al. found that most of the students analyzed were of normal weight; meanwhile, only half of the students recognized this fact. The disturbed body image mainly concerned girls, as 17% perceived themselves as overweight, while it was found in half of these respondents [20].

Another factor that predisposes to eating disorders is devoting too much time and thought to food. It is observed that nowadays high-calorie products are being advertised, and there is a trend for eating a variety of foods without limits. In turn, it is also fashionable to have an athletic figure, which is associated with achieving success [9]. These two contradictory trends may influence the public to use various types of diets to lose weight in a short period. In addition, physical activity is observed to burn off excessive calories eaten and thus maintain a slim figure. All these factors can lead to an emerging obsession with food [21]. Especially according to some studies among nutrition students, who experience pressure from society to eat properly and have a slim figure. This can contribute to excessive control over the food they eat [22]. Of the dieters surveyed, 28% marked the answer “always”, “usually”, or “often” when asked about spending too much time and thoughts about food, while the other group had a lower percentage. at 21%; *p* = 0.004. In contrast, in a study by Taha et al. a similar percentage (33.5%) of students admitted that thoughts about food absorb them a lot [23].

One of the hallmarks of eating disorders is a paralyzing fear of gaining weight. Among the dietetics students surveyed, almost half—43%—are “always”, “usually”, or “often” terrified at the thought of being obese, and among students in other majors this percentage is slightly lower at 39% (*p* > 0.05-NS). On the other hand, among nutrition students, despite being of normal weight or even underweight, 28% are consumed with the desire to be thinner, and among students in other majors, the percentage is 27%; *p* > 0.05-NS. Furthermore, 27% of normal-weight or underweight dietetics students “always,” “usually”, or “often” think insistently about fat on their bodies. In contrast, the percentage in the second group is lower at −16%; *p* < 0.05. Contrasting these results with a study of Brazilian adolescents, by far the majority of respondents declared fear of weight gain [24]. In a study by Toral et al., 26.7% of nutrition students were observed to have significant dissatisfaction regarding their body image [25].

For people with eating disorders, especially among anorexia patients, every meal is a certain ritual. Slimming people know the caloric value of the foods they eat. The marker of lifelong success becomes counting the calories of meals every day, as well as those burned during exercise [26]. Among nutrition students, the majority of respondents (79%) are “always,” “usually”, or “often” aware of the caloric value of the products they eat. In contrast, among non-diet students, almost half as many—41%—know exactly how many calories a product has; *p* = 0.000. Additionally, 39% of nutrition students think about burning calories during exercise. In contrast, 37% of the other group thinks about calories lost during physical activity; *p* > 0.05-NS. In contrast, in a study by other authors, among Mexican nutrition students, almost half of the respondents who had symptoms of an appetite disorder counted the caloric value before meals [27], while among students in Saudi Arabia, 67% of students thought about calories burned during exercise [22]. However, it is worth noting that caloric awareness of products may not represent a real risk of anorexia among dieters, among others, as it may be characteristic of the nutrition students studied due to their field of study.

According to the accepted scores from the EAT-26 test, Part A, of the dietetics students surveyed, 15% scored 20 points or more, which may indicate concern about diet, weight, or problematic behavior. In the second group of subjects, the result was the same —15% of students in other majors also received a positive test result (EAT ≥ 20), which may indicate a risk for eating disorders. This percentage was comparable to a previous study in Florida in the United States in which 10% of nutrition students, as well as the same percentage of students in other majors, received a score of −20 or higher on the EAT-26 test. Compared to other countries, this percentage is lower than that of French students (20.5%) [18] and medical students from Pakistan (22.75%) [28].

In Part B of the survey on behavioral questions, 33% of dietetics students met a criterion that could indicate a risk of developing an appetite disorder. These were behaviors, occurring at a certain time of the type: paroxysmal overeating, vomiting, use of diuretics, laxatives, excessive exercise, or loss of 10 kg recently. In the second group of subjects, the result was slightly lower at 28% (*p* > 0.05-NS), suggesting the need for further investigation into eating disorders. In a study by Harris et al., there were also no significant differences between the two groups regarding behavioral questions [29]. In contrast, in a study among students in Palestine using the EAT-26 test, slightly more—46%—of students endorsed at least one of the additional behavioral behaviors associated with an eating disorder [30].

In contrast, according to the BMI thresholds presented in the EAT-26 interpretation, 14% of dietetics students were underweight compared to their age and gender, while among students in other majors this problem affected slightly more—17% of respondents. The study by Zhiping et al. also found no differences in underweight between the two groups. The percentages, however, were slightly smaller, as 4.1% of nutrition students were underweight compared to age norms, while students in other majors were underweight by 5.6% [17].

Based on the overall accepted scores and interpretation of the EAT-26 test, as many as almost half (46%) of dietetics students, and as it turned out, the same percentage of students in other majors, met at least one criterion that could indicate a risk of developing an eating disorder. As can be seen, the study did not find that dietetics students are at higher risk of developing an eating disorder than students in other majors. Other studies have also found similar results. Among Portuguese students aged 18–15, there was no difference in the risk of developing eating disorders between dietetics students and other non-food majors [19]. In a study in Washington, using the EAT-26 test, no such relationship was observed either [31].

It is becoming a common belief that nutrition students are more predisposed to appetite disorders than students in other majors. The implication is that future dietitians may feel pressure from their surroundings to eat healthily and present an impeccable figure at all times. In addition, some patients might not use the services of a dietitian with excessive body weight, which could indicate that person’s lack of competence. In addition, nutrition students are “fed” during classes with constant messages about proper nutrition. All of these factors can give rise to an obsession with food, proper eating habits, and outward appearance [32,33]. However, in our work, we did not find a correlation (according to the accepted EAT-26 test scores) that dietetics students are more likely to have eating disorders than non-dietetics students. This may indicate that not only in nutrition students, but also in students of other majors, the problem of an abnormal relationship with food is becoming common. In addition, it is worth noting that the survey among non-nutrition students was conducted electronically, where often the social media group at a given university of 2,000 people. The available survey likely attracted the attention of students who observed some abnormal eating behavior in themselves or were interested in this topic.

Although according to the final accepted EAT-26 test scores, there was no correlation that dietetics students were more predisposed to eating disorders, such correlations appeared in individual responses. A significant influence of the type of field of study on too frequent thoughts about body fat, despite being of the right weight or even underweight, preoccupation with food, and devoting too much time and thought to food was noted. These symptoms can predispose to the onset of eating disorders. Moreover, a significant effect of the type of college on losing 10 kg in the past 6 months was noted (*p* = 0.006), as well as past treatment for eating disorders (*p* < 0.05). A significant effect of the type of college was also observed on the caloric awareness of meals eaten (*p* = 0.000), consumption of diet foods (*p* = 0.000), avoidance of sugar-containing products (*p* = 0.142), or eating for longer periods than others (*p* = 0.000). Although eating diet foods or avoiding sweets may indicate a restrictive approach to eating and eating a meal at a longer time than others may be associated with anorexia, these behaviors may not, however, represent a real threat of an eating disorder. They are likely to be characteristic of the dietetics students surveyed due to the type of study and may also be indicative of normal eating habits.

In addition, it should be emphasized that even though a significant proportion of students (46%) were noted to meet at least one criterion in the EAT-26 test that may indicate a risk of developing an appetite disorder, this does not necessarily mean a real risk of bulimia or anorexia among the surveyed students. It is also worth mentioning that the presence of a single symptom characteristic of anorexia nervosa or gluttonousness, even if it appears sporadically in the company of other symptoms of the disease, should not be rigidly associated with the appearance of anorexia or bulimia nervosa in a person. Despite appearances, eating disorders are difficult to diagnose. Some symptoms in different types of eating disorders overlap. This makes it difficult to correctly diagnose the disease. It is important to catch certain tendencies at an early stage that may indicate a risk of developing an eating disorder in the future. Thus, it is urged that as many screening tests as possible be conducted in this direction among young people to detect a possible disease at an early stage and implement appropriate treatment before it is too late.

## 5. Conclusions

Almost half of the respondents in both study groups showed a predisposition to eating disorders based on the EAT-26 test. Despite the presence of a correlation in individual responses that dietetics students are more predisposed to eating disorders, no such relationship was found according to the final EAT-26 test scores. However, it was observed that non-dietetics students who had low BMI values showed higher tendencies toward behaviors indicative of eating disorders.

## Figures and Tables

**Table 1 nutrients-14-03210-t001:** BMI classification for adults [10].

BMI (kg/m^2^)	Interpretation of BMI
<16.00	Starvation
16.00–16.99	Emaciation
17.00–18.49	Underweight
18.50–24.99	Body weight normal
25.00–29.99	Overweight
30.00–34.99	First-degree obesity
35.00–39.99	Grade II obesity
≥40.00	Grade III obesity

**Table 2 nutrients-14-03210-t002:** Scoring on behavioral questions [11].

	Never	1/Month	2–3/Month	1/Week	2–6/Week	1/Day
**(A) Overeating**	-	-	X	X	X	X
**(B) Vomiting**	-	X	X	X	X	X
**(C) Pharmacology**	-	X	X	X	X	X
**(D) Exercises**	-	-	-	-	-	X
**(E) Weight-loss**	-	X	-	-	X	-

**Table 3 nutrients-14-03210-t003:** Interpretation of BMI compared to age and gender norms [11].

Age	9	10	11	12	13	14	15	16	17	18	19	20	>20
**BMI-female**	14.0	14.5	14.5	15.0	15.5	16.0	16.5	17.0	17.5	18.0	18.0	18.5	19.0
**BMI-male**	14.0	14.5	15.0	15.0	16.0	16.5	17.0	17.5	18.0	18.5	19.0	19.5	20.5

**Table 4 nutrients-14-03210-t004:** Summary of eating disorder risk estimation (EAT-26) (*n* = 310).

EAT-26	Dietetics Students		Other Students		*p*-Value
	Elevated Risk	No Risk	Elevated Risk	No Risk	
**Part A**	15%	85%	15%	85%	*p* > 0.05
**Part B**	33%	67%	28%	72%	
**Entire**	46%	54%	46%	54%	
**All adjusted for BMI**	**12%**	88%	**31%**	69%	** *p* ** ** = 0.001**

## Data Availability

The original contributions presented in the study are included in the article; further inquiries can be directed to the corresponding author.

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
