# Peer review of "Comparison of the Prevalence of Eating Disorders among Dietetics Students and Students of Other Fields of Study at Selected Universities (Silesia, Poland)"

_nutrients, 2022, doi:10.3390/nu14153210_

Round 1

Reviewer 1 Report

This paper compares the prevalence of eating disorders among dietetics students and students of other major fields of study at some selected universities. The study uses the questionnaire method to obtain scores of different items and analyze the scores to compare two groups. I have some comments.

1.     In the dietetics student group, why are the female (96%) and male students (4%) so imbalanced?

2.     Does the first-degree student mean the undergraduate students?

3.     Line 33. relation-ship relationship

4.     Lines 69-70. “Anorexia nervosa is increasingly being observed, including among people who train recreationally.” Could you explain more about this or include a citation?

5.     Lines 95-97. Are the number of respondents in different types of the university the same for both groups?  

6.     Line 129. There is more than one author. Why do you use “I”?

7.     Line 140. What did you mean “December 5, 1996”?

8.     Line 143. You do not need to mention this “The obtained results were transported to Microsoft Excel and then processed graphically.”. Since you have mentioned it, I suggest including this excel file in the supplementary materials.

9.     Lines 177-179. “ which may indicate the risk of developing an eating disorder

(p>0.05-NS)”. It is not clear.

10.  The questionnaire EAT-26 can be included as a supplementary file. 

Author Response

Dear Reviewer,

Thank you very much for the time and tremendous amount of work you put into reviewing our manuscript. We have done our best to respond to all the suggestions indicated in your review. All changes have been marked in red in the text.

Re 1. The disproportion is due to the fact that in Poland dietetics is a virtually all-female field of study. It is rarely chosen by men. Therefore, in further analysis it was not decided to take gender as a predisposing factor. Although studies indicate that eating disorders are more likely to affect the female gender, but in the current project such results would be heavily skewed.

Re 2. Yes, the paper corrected that it is about bachelor.

Re 3. Corrected.

Re 4. The passage was removed because it was indeed a far-fetched abbreviation of thought and could introduce doubt.

Re 5. 155 dietetics students and 155 students of other majors participated in the study. In this respect, the groups were equal. The group of students from other universities included: 51 economic science students, 52 science students, 52 music students.

Re 6. Corrected.

Re 7. This is an indication of the date from which the Polish Law that talks about conducting medical experiments originates. Before the date is the name of this law. It is a standard procedure used in research to point to the legal act talking about keeping the research in proper ethical order.

Re 8. The unnecessary sentence has been removed.

Re 9. Corrected as suggested.

Re 10. p>0.05-NS means no statistical significance. The abbreviation has already been explained in the methodological part of the study.

Re 11. The questionnaire has been added.

Thank you again and best regards.

Reviewer 2 Report

The comments are in a document  attached

Author Response

Dear Reviewer,

Thank you very much for the time and tremendous amount of work you put into reviewing our manuscript. We have done our best to respond to all the suggestions indicated in your review. All changes have been marked in red in the text.

Re 1. A paragraph was added in the introduction regarding the lack of clear data on the spread of eating disorders and the lack of data that related to the topic of the study. The information on the elderly population was removed, but it was decided to leave the information on physically active people, as this is one of the factors that alludes to the issue of caring about diet and taking care of the body, which are important in this section of the work. The excerpt from the discussion pointed out in the review eventually made its way into the introduction. In addition, a paragraph on the project itself - its relevance to the field - was added. The research stages for the research hypothesis were corrected.

Re 2. The study emphasized that risks were studied (the passage was marked).

Re 3. The unnecessary passage was removed from the discussion.

Thank you again and best regards.

Reviewer 3 Report

This paper examines whether there was a difference in the risk of eating disorders between students who have been selected for dietetics and those who have not.

Although the topic is interesting, as noted in the authors' Discussion, there is already a large body of previous research on this topic.

Major Issues

1. please provide a brief summary of what is known and what is not known from previous studies in the Introduction, and describe the novelty and significance of this study.

2. The Discussion section is full of information, but there is no need for an excessive review of the previous literature.

3. as stated by the authors, it is possible that students who are ahead of their dietetics studies may have an inherent tendency to develop eating disorders. Were there any questions in the survey that would adjust for this?

For example, did you check to see if they had ever visited a hospital or been diagnosed with an eating disorder?

4. Regarding statistical methods, it seems to me that there were more appropriate methods such as multivariate analysis.

  I would like to see significant improvement.

Author Response

Dear Reviewer,

Thank you very much for the time and tremendous amount of work you put into reviewing our manuscript. We have done our best to respond to all the suggestions indicated in your review. All changes have been marked in red in the text.

Re 1. A paragraph was added in the introduction regarding the lack of clear data on the spread of eating disorders and the lack of data that related to the topic of the study. In addition, a paragraph was added regarding the project itself - its relevance to the field. The research stages for the research hypothesis were corrected.

Re 2. Two paragraphs that actually did not fit the nature of the study were removed.

Re 3. The study used the EAT-26 questionnaire plus metric questions (to identify sociodemographic characteristics of the study participant), but prior to the study, an interview assessing mental health history was conducted (added this information to the paper).

Re 4. The chi-square test was used for the analyses, as the data obtained were non-parametric and similar analyses were also used by other authors.

Thank you again and best regards.

Round 2

Reviewer 1 Report

The authors have revised the paper according to my comments.

Reviewer 3 Report

no further comments